# The impact of cardiopulmonary exercise-derived scoring on prediction of cardio-cerebral outcome in hypertrophic cardiomyopathy

Jae-Man Lee[1◉], Hyun-Bin Park[1◉], Jin-Eun Song[1◉], In-Cheol Kim[1]*, Ji-Hun Song[1], Hyungseop Kim[1], Jaewon Oh[2], Jong-Chan Youn[3], Geu-Ru Hong[2], Seok-Min Kang[2]

1 Division of Cardiology, Department of Internal Medicine, Cardiovascular Center, Keimyung University Dongsan Hospital, Keimyung University College of Medicine, Daegu, Republic of Korea, 2 Division of Cardiology, Severance Cardiovascular Hospital, Yonsei University College of Medicine, Seoul, Republic of Korea, 3 Division of Cardiology, Department of Internal Medicine, Seoul St. Mary's Hospital, College of Medicine, The Catholic University of Korea, Seoul, Republic of Korea

◉ These authors contributed equally to this work.
* kimic@dsmc.or.kr

**Data Availability Statement:** All relevant data are within the manuscript and its Supporting information files.

## Abstract

### Background

Sudden cardiac death (SCD) and stroke-related events accompanied by atrial fibrillation (AF) can affect morbidity and mortality in hypertrophic cardiomyopathy (HCM). This study sought to evaluate a scoring system predicting cardio-cerebral events in HCM patients using cardiopulmonary exercise testing (CPET).

### Methods

We investigated the role of a previous prediction model based on CPET, the HYPertrophic Exercise-derived Risk score for Heart Failure-related events (HyperHF), which is derived from peak circulatory power ventilatory efficiency and left atrial diameter (LAD), for predicting a composite of SCD-related (SCD, serious ventricular arrhythmia, death from cardiac cause, heart failure admission) and stroke-related (new-onset AF, acute stroke) events. The Novel HyperHF risk model using left atrial volume index (LAVI) instead of LAD was proposed and compared with the previous HCM Risk-SCD model.

### Results

A total of 295 consecutive HCM patients (age 59.9±13.2, 71.2% male) who underwent CPET was included in the present study. During a median follow-up of 742 days (interquartile range 384–1047 days), 29 patients (9.8%) experienced an event (SCD-related event: 14 patients (4.7%); stroke-related event: 17 patients (5.8%)). The previous model for SCD risk score showed fair prediction ability (AUC of HCM Risk-SCD 0.670, p = 0.002; AUC of HyperHF 0.691, p = 0.001). However, the prediction power of Novel HyperHF showed the highest value among the models (AUC of Novel HyperHF 0.717, p<0.001).

**Funding:** This research was supported by the Bisa Research Grant of Keimyung University in 2017.

**Competing interests:** The authors have declared that no competing interests exist.

## Conclusions

Both conventional HCM Risk-SCD score and CPET-derived HyperHF score were useful for prediction of overall risk of SCD-related and stroke-related events in HCM. Novel HyperHF score using LAVI could be utilized for a better prediction power.

## Introduction

The most hazardous complication of hypertrophic cardiomyopathy (HCM) is sudden cardiac death (SCD), with a prevalence of 0.2% per year [1–3]. However, atrial fibrillation (AF), which is the most common sustained arrhythmia affecting 20% of patients with HCM, can progress to serious neurologic complications resulting in deterioration of quality of life and increased mortality [4–7].

Stratification of the risk for HCM complications is crucial to the section of an appropriate therapeutic strategy. The most widely used risk model is the HCM Risk-SCD score, which was developed by the European Society of Cardiology. The model provides individualized 5-year risk estimates using major risk factors of maximal wall thickness, left atrial diameter, maximal left ventricular outflow tract pressure gradient, family history of SCD, non-sustained ventricular tachycardia (NSVT), unexplained syncope history, and age at clinical evaluation. The model predicted SCD in HCM patients with high accuracy [8,9].

Cardiopulmonary exercise testing (CPET) has been suggested as a useful approach to assess objective functional capacity as well as risk stratification in HCM patients. In the 2020 AHA HCM guideline, CPET was included in risk stratification aiding cardiologists to quantify the degree of functional limitation and select patients for heart transplantation or mechanical circulatory support [10]. Previous studies suggested a possible role of CPET assessment in stratifying overall HCM prognosis, thrombo-embolic risk, and SCD risk [11–17]. Moreover, CPET can also provide a non-invasive method for assessing the cardiovascular, pulmonary and skeletal muscle components of exercise performance [10]. Based on these advantages, a CPET-derived risk model, the HYPertrophic Exercise-derived Risk HF (HyperHF) score, which includes both CPET and echocardiographic parameters, was suggested as a useful predictor for SCD-related events [17]. However, there is no available scoring system that provides integrated risk prediction of both SCD-related and stroke-related events. Furthermore, there is adequate data reporting left atrial volume index (LAVI) to be a better measure of left atrial dilatation as compared to left atrial diameter (LAD), providing a more accurate assessment of left atrial size than conventional M-mode LAD. Hence, we created Novel HyperHF scoring system, a CPET-derived risk model that uses LAVI instead of LAD for measuring left atrial size. The aim of this study was to test the risk stratification ability of Novel HyperHF score. By comparing it with previous risk scoring models, we could propose a superior risk stratification model in predicting both SCD-related and stroke-related events.

## Materials and methods

### Study population

A total of 330 consecutive patients with true HCM who underwent CPET were recruited and prospectively followed in HCM centers from two tertiary University hospitals in the Republic of Korea between November 2011 and May 2018. Patients aged younger than 18 years (n = 2) or diagnosed with left ventricular hypertrophy (n = 8) were excluded to select true adult HCM

patients. Additionally, patients with atrial fibrillation (n = 18), infiltrative cardiomyopathy (n = 3), pacemaker rhythm (n = 1), history of septal myectomy (n = 2) and history of alcohol septal ablation (n = 1) were excluded. A total of 295 patients were finally enrolled in this study, all with normal sinus rhythm without history of atrial fibrillation (Fig 1). The diagnosis of HCM was based on a maximal wall thickness ≥15 mm unexplained by abnormal loading conditions or in accordance with published criteria for diagnosis of disease in relatives of patients with unequivocal disease [9,10]. Sub-types of enrolled HCM patients were obstructive (n = 56), non-obstructive (n = 113) and apical (n = 126). This study conforms to the ethical guidelines of the Declaration of Helsinki and was approved by the institutional ethics board of Yonsei University Severance Hospital (no. 4-2015-0264). Written informed consent was waived because of the retrospective nature of this study.

## Transthoracic echocardiography

Complete transthoracic echocardiography (TTE) was performed in all patients using commercially available scanners (GE Vivid E9, GE Healthcare, Waukesha, WI, USA; Philips IE33, Philips Medical Systems, Andover, MA, USA; Siemens Sequoia C512, Siemens Medical Solutions, Mountain View, CA, USA), including 2D, pulsed-wave, continuous-wave and color Doppler imaging. All studies were performed at rest in the left lateral position. Left ventricular (LV) dimensions and LAD were measured with M-mode in the parasternal short-axis view but the most thickened segment was evaluated throughout the examination. The LAVI and LV ejection fraction (EF) were measured using the modified Simpson's method from images with apical two- and four-chamber views. Continuous-wave Doppler was used to assess aortic outflow peak velocity as well as peak acceleration velocity where it was present. Valsalva maneuver was additively applied when available. The E/e' ratio was calculated based on the mitral E velocity obtained using pulsed-wave Doppler, and the mitral annular e' velocity at the interventricular septal annulus was obtained using tissue Doppler imaging.

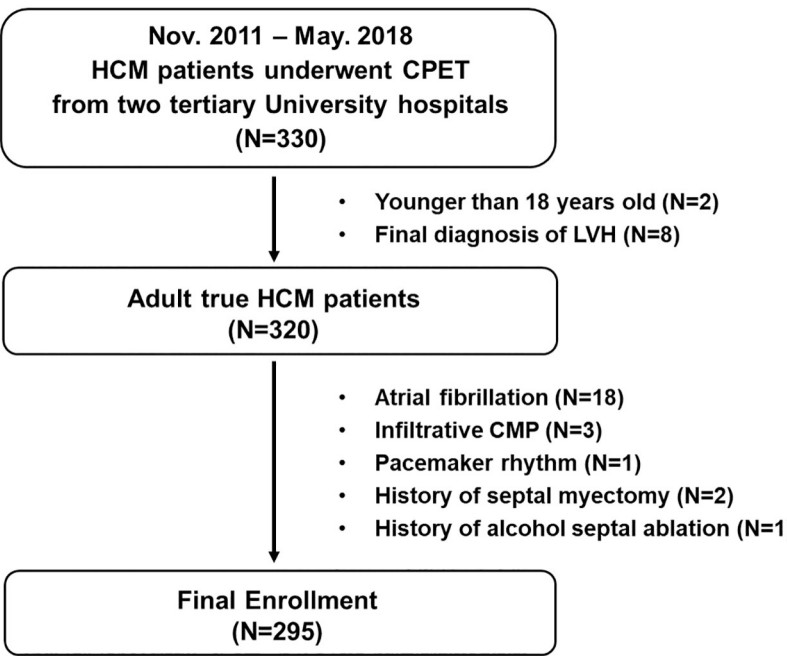

**Fig 1. Flow chart showing selection of the study population.** (HCM, hypertrophic cardiomyopathy; CPET, cardiopulmonary exercising test; LVH, left ventricular hypertrophy; CMP, cardiomyopathy).

## Cardiopulmonary exercise testing and related parameters

A CPET was performed on a treadmill according to the modified Bruce ramp protocol. Patients were strongly encouraged to achieve a peak respiratory exchange ratio (RER) >1.10. Expired gases were collected continuously throughout exercise and analyzed for ventilator volume, oxygen ($O_2$) content, and carbon dioxide ($CO_2$) content using a calibrated metabolic cart (Quark CPET, COSMED, Chicago, IL, USA). Expired gases were measured every 15 seconds. During the exercise test, monitoring consisted of continuous 12-lead electrocardiography, manual blood pressure (BP) measurements and heart rate recordings at every stage via the ECG. CPET was terminated based on the following criteria: patient request, ventricular tachycardia, horizontal or down-sloping ST segment depression of ≥2 mm, or a drop in systolic BP ≥20 mm Hg during exercise. A qualified exercise physiologist conducted each test, under supervision of a physician.

The following variables were derived from the CPET results: peak $VO_2$; peak RER, defined by the ratio of $CO_2$ production to $O_2$ consumption at peak effort; and the minute ventilation–carbon dioxide production ($VE/VCO_2$) slope, defined as the slope of the increase in peak ventilation/increase in $CO_2$ production throughout exercise. Peak RER had the highest 30s average value during the last stage of the test. Heart rate reserve is defined as the difference between basal and peak heart rate.

## Clinical outcomes

Clinical outcomes were evaluated during a median follow-up of 742 days (interquartile range 384–1047 years). SCD-related events comprised cardiac death, symptomatic ventricular tachycardia or ventricular fibrillation, and admission due to heart failure aggravation. Stroke-related events consisted of new-onset atrial fibrillation and acute stroke after enrollment. Overall events included both SCD-related events and stroke-related events.

## Risk model verification

In HCM Risk-SCD, the risk of SCD in 5 years for an individual HCM patient can be calculated from the following equation: $P^{SCD}$ at 5 years = $1 - 0.998^{exp (Prognostic Index)}$, where Prognostic Index = $0.15939858^*$Maximal wall thickness (mm)$- 0.00294271^*$Maximal wall thickness$^2$ (mm$^2$) + $0.0259082^*$ Left atrial diameter (mm) + $0.00446131^*$Maximal left ventricular outflow tract gradient (mmHg) + $0.4583082^*$Family history SCD + $0.82639195^*$NSVT + $0.71650361^*$Unexplained syncope—$0.01799934^*$Age at clinical evaluation (years).

HyperHF score indicates the probability of any HF-related event over five years for a single patient and is calculated as $P^{HFevents-at-5 years} = 1 - 0.910^{exp(Index)}$ where 0.910 is the survival probability at five years and index is the sum of the products of the (centered and scaled) covariates and their coefficients estimated via the Cox model [Index = $0.045^*$LAD (mm) $-0.000285^*$pVO$_2$ CP (% of predicted) + $0.071^*$ VE/VCO$_2$ slope].

To overcome the limitations of LAD compared with LAVI, Novel HyperHF score including LAVI instead of LAD was compared with the previous two risk models (HCM Risk-SCD, HyperHF).

## Statistical analysis

Statistical analysis was performed using SPSS software version 20 (IBM SPSS Statistics for Windows, IBM Corp., Armonk, NY, USA). Unless otherwise indicated, all data of continuous variables are presented as mean ± standard deviation and were compared with an independent *t*-test and Pearson's correlation coefficient analysis, as appropriate. Variables that were non-

normally distributed were compared using the Mann-Whitney U-test. Categorical variables were compared using the chi-square test or Fisher's exact test, as appropriate. Simple and multivariable linear regression was applied to evaluate the significance of variables. Tests of the proportional hazards assumption for each covariate were obtained with the Kaplan-Meier estimate of survival distribution. A receiver-operating characteristic (ROC) analysis was considered to determine the predictive capability of each risk model in identifying the HF endpoint. A P-value ≤0.05 was considered statistically significant.

## Results

### Baseline characteristics of the study population

Baseline characteristics including patient characteristics are displayed in Table 1 according to presence and absence of overall events. There was no significant difference between the two groups.

Echocardiographic parameters are shown in Table 2. Ejection fraction and left ventricular size were similar in the two groups. Left atrial size (LAD, LAVI) was larger in the event (+) group with thicker myocardium, p<0.001;). Compared to the event (-) group, event (+) group showed larger LA size (LAD 44.7 ± 5.1 vs. 40.8 ± 6.0 mm, p = 0.001; LAVI 46.0 ± 14.9 vs. 35.7 ± 13.5 mL/m$^2$), thicker myocardium (maximum wall thickness 21.7 ± 5.4 vs. 19.2 ± 4.3 mm, p = 0.005), shorter deceleration time (183.3 ±63.3 vs. 205.9 ± 46.4, p = 0.017), and lower mitral annular tissue velocity (s' 5.70 ± 1.84 vs. 6.53 ± 1.29, p<0.001; a' 6.30 ± 1.79 vs. 7.65 ± 1.85, p = 0.006).

Table 3 summarizes the CPET parameters. Patients with event showed significantly lower peak VO$_2$ (23.1 ± 5.7 vs. 27.3 ± 6.4 mL/kg/min, p = 0.001) with higher VE/VCO$_2$ slope (32.5 ± 4.7 vs. 29.7 ± 3.9 mL/kg/min, p<0.001).

### Study endpoints

Overall study endpoints are shown in Table 4. During a median follow-up of 742 days (interquartile range 384–1047 years), 29 patients (9.8%) experienced at least one event. There were 14 patients (4.7%) with SCD-related events, and 17 patients (5.8%) with stroke-related events.

### Model verification

Area under the curve (AUC) values were compared for HCM Risk-SCD score, HyperHF score and Novel HyperHF score in predicting overall events using the ROC curve. HyperHF score showed numerically higher AUC value compared with HCM Risk-SCD score (0.697 vs. 0.670). However, Novel HyperHF score showed the highest value (0.717) (Fig 2).

For SCD-related events, the highest AUC value was noted for HCM Risk-SCD score (AUC 0.715, p = 0.007), followed by Hyper HF score (AUC 0.695, p = 0.014) and Novel HyperHF score (AUC 0.692, p = 0.015) (S1 Fig). For stroke-related events, Novel HyperHF score showed the highest AUC value with statistical significance (AUC 0.659, p = 0.028) (S2 Fig). However, other risk prediction models did not show statistical significance (Hyper HF AUC 0.620, p = 0.098; HCM Risk-SCD AUC 0.607, p = 0.138).

Results of sensitivity, specificity, positive predictive value, and negative predictive value are displayed for overall events, SCD-related events, and stroke-related events in S1–S3 Tables, respectively. Kaplan-Meier curve for event-free survival showed significant discrimination between two groups of Novel HyperHF score < 4.5% versus ≥ 4.5% (Fig 3).

**Table 1. Baseline characteristics of overall HCM patients and patients with positive events vs. negative events (*: P value for Event (+) vs. Event (-)).**

| | Overall HCM (n = 295) | Event (+) (n = 29) | Event (−) (n = 266) | p value* |
|---|---|---|---|---|
| Age (years) | 53.9 ± 13.2 | 53.9 ± 13.4 | 53.9 ± 13.2 | 0.996 |
| Male sex, n (%) | 210 (71.2) | 23 (79.3) | 187 (70.3) | 0.277 |
| Body mass index, kg/m$^2$ | 24.7 ± 3.1 | 24.1 ± 3.5 | 24.8 ± 3.0 | 0.262 |
| Hypertension, n (%) | 134 (45.4) | 8 (27.6) | 126 (47.4) | 0.034 |
| Diabetes mellitus, n (%) | 48 (16.3) | 3 (10.3) | 45 (16.9) | 0.296 |
| Dyslipidemia, n (%) | 102 (34.6) | 8 (27.6) | 94 (35.3) | 0.392 |
| Hemoglobin, mg/dL | 14.8 ± 1.6 | 15.2 ± 1.2 | 14.8 ± 1.6 | 0.129 |
| Creatinine, mg/dL | 0.91 ± 0.33 | 0.94 ± 0.24 | 0.90 ± 0.33 | 0.589 |
| Antiplatelet, n (%) | 92 (31.2) | 9 (31.0) | 83 (31.2) | 0.985 |
| Beta-blocker, n (%) | 120 (40.7) | 13 (44.8) | 107 (40.2) | 0.633 |
| ACEi, n (%) | 10 (3.4) | 10 (100) | 0 (0) | 0.288 |
| ARB, n (%) | 100(31.2) | 8 (27.6) | 92 (34.6) | 0.450 |
| Diuretics, n (%) | 20 (6.8) | 4 (13.8) | 16 (6.0) | 0.253 |
| Statins, n (%) | 82 (27.8) | 10 (34.5) | 72 (27.1) | 0.399 |

(ACEI, angiotensin-converting enzyme inhibitor; ARB, angiotensin-receptor blocker).

**Table 2. Echocardiography data of overall HCM patients and patients with positive events vs. negative events (*: P value for Event (+) vs. Event (-)).**

| | Overall HCM (n = 295) | Event (+) (n = 29) | Event (−) (n = 266) | p value* |
|---|---|---|---|---|
| **Ejection fraction, %** | 69.0 ± 7.5 | 67.2 ± 12.0 | 69.2 ± 6.8 | 0.382 |
| **LVEDD, mm** | 47.5 ± 5.0 | 48.0 ± 6.2 | 47.4 ± 4.9 | 0.536 |
| **LVESD, mm** | 29.9 ± 4.4 | 30.9 ± 6.4 | 29.8 ± 4.1 | 0.383 |
| **LAD, mm** | 41.2 ± 6.0 | 44.7 ± 5.1 | 40.8 ± 6.0 | 0.001 |
| **IVSd, mm** | 14.6 ± 5.1 | 16.6 ± 6.2 | 14.6 ± 4.9 | 0.105 |
| **PWDd, mm** | 10.5 ± 2.4 | 11.0 ± 3.0 | 10.4 ± 2.3 | 0.265 |
| **LVMI, mm** | 124.4 ± 51.8 | 166.9 ± 107.3 | 119.1 ± 38.8 | 0.285 |
| **LA volume index, ml/m$^2$** | 36.7 ± 13.9 | 46.0 ± 14.9 | 35.7 ± 13.5 | <0.001 |
| **Maximum thickness, mm** | 19.4 ± 4.5 | 21.7 ± 5.4 | 19.2 ± 4.3 | 0.005 |
| **E velocity, m/s** | 0.63 ± 0.17 | 0.60 ± 0.19 | 0.62 ± 0.16 | 0.435 |
| **A velocity, m/s** | 0.64 ± 0.22 | 0.56 ± 0.16 | 0.65 ± 0.23 | 0.086 |
| **E over A** | 1.12 ± 0.60 | 1.15 ± 0.39 | 1.12 ± 0.62 | 0.056 |
| **Deceleration time, ms** | 203.7 ± 48.7 | 183.3 ± 63.3 | 205.9 ± 46.4 | 0.017 |
| **s', cm/s** | 6.44 ± 1.37 | 5.70 ± 1.84 | 6.53 ± 1.29 | 0.001 |
| **e', cm/s** | 4.80 ± 1.78 | 4.37 ± 1.64 | 4.85 ± 1.80 | 0.315 |
| **a', cm/s** | 7.50 ± 1.88 | 6.30 ± 1.79 | 7.65 ± 1.85 | 0.006 |
| **E/e'** | 13.86 ± 5.5 | 14.03 ± 6.5 | 13.84 ± 5.38 | 0.858 |

(LAD, left atrial anterior-posterior dimension; LVEDD, left ventricular end-diastolic dimension; LVESD, left ventricular end systolic dimension; IVSd, interventricular septum thickness at end-diastole; PWDd, posterior wall thickness at end-diastole; LVMI, left ventricular mass index; LAVI, left atrial volume index; E, peak velocity of early diastolic trans-mitral flow; A, peak velocity of late trans-mitral flow; s', peak velocity of systolic mitral annular motion as determined by pulsed wave Doppler; e', peak velocity of early diastolic mitral annular motion as determined by pulse wave Doppler; a', peak velocity of diastolic mitral annular motion as determined by pulsed wave Doppler).

**Table 3. Cardiopulmonary exercise testing data of overall HCM patients and patients with positive events vs. negative events (\*: P value for Event (+) vs. Event (-)).**

| | Overall HCM | Event (+) | Event (−) | p value* |
|---|---|---|---|---|
| | (n = 295) | (n = 29) | (n = 266) | |
| Exercise duration, sec | 735.2 ± 233.1 | 659.5 ± 202.4 | 743.5 ± 235.1 | 0.065 |
| Peak VO$_2$, mL/kg/min | 26.9 ± 6.4 | 23.1 ± 5.7 | 27.3 ± 6.4 | 0.001 |
| VE/VCO$_2$ slope | 30.0 ± 4.1 | 32.5 ± 4.7 | 29.7 ± 3.9 | <0.001 |
| LT time, sec | 415.5 ± 198.6 | 352.1 ± 179.0 | 422.4 ± 199.7 | 0.070 |
| METs | 7.68 ± 1.84 | 6.60 ± 1.62 | 7.80 ± 1.82 | 0.001 |
| Peak RER | 1.15 ± 0.22 | 1.10 ± 0.10 | 1.16 ± 0.23 | 0.171 |
| VD/VTp | 0.28 ± 0.03 | 0.30 ± 0.03 | 0.28 ± 0.03 | 0.017 |
| P$_{ET}$CO$_2$, mmHg | 38.6 ± 6.1 | 36.0 ± 5.6 | 38.9 ± 6.1 | 0.014 |
| OUES | 2235.7 ± 718.7 | 2008.7 ± 709.0 | 2259.8 ± 716.9 | 0.090 |
| Baseline SBP, mmHg | 123.2 ± 16.7 | 124.6 ± 20.6 | 123.1 ± 16.3 | 0.694 |
| Peak SBP, mmHg | 184.3 ± 35.5 | 171. 8 ± 38.6 | 185.6 ± 34.9 | 0.047 |
| Baseline HR, bpm | 67.8 ± 11.5 | 64.4 ± 10.5 | 68.2 ± 11.6 | 0.092 |
| Peak HR, bpm | 147.8 ± 25.4 | 132.4 ± 26.7 | 149.5 ± 24.7 | 0.001 |
| HRR, bpm | 78.4 ± 24.8 | 68.0 ± 23.5 | 79.5 ± 24.7 | 0.017 |

(VO$_2$, oxygen consumption, VE/VCO2 slope, relation between ventilation vs. carbon dioxide production; LT time, lactate threshold time; METs, metabolic equivalent; RER, respiratory exchange ratio; VD/VTp, peak ratio of dead space to tidal volume; OUES, oxygen uptake efficiency slope; SBP, systolic blood pressure; HRR, heart rate recovery).

To demonstrate that the novel HyperHF score is responsible for risk prediction regardless of conventional risk factors, multivariate analysis was performed. As a result, Novel HyperHF model was significantly related with the outcome. (Table 5).

## Discussion

The main findings of the current study are as follows: 1) both conventional HCM Risk-SCD score and CPET-derived HyperHF score were useful for prediction of overall risk of SCD-related and stroke-related events in HCM; 2) Novel HyperHF score using LAVI could be utilized for a better prediction power; 3) Novel HyperHF ≥ 4.5% showed significantly poor outcome compared to those with Novel HyperHF < 4.5%.

Even though two previous risk models (HCM Risk-SCD score and HyperHF score) were originally developed to predict SCD risk of HCM patients, components of these models are related to AF and acute stroke. Therefore, we tested whether this widely used risk model can

**Table 4. SCD-related and stroke-related events of the study population.**

| Events | Number |
|---|---|
| SCD-related events, n (%) | 14 (4.7) |
| Cardiac death, n (%) | 1 (0.3) |
| Admission due to HF aggravation, n (%) | 2 (0.7) |
| Symptomatic ventricular arrhythmia, n (%) | 11 (3.7) |
| Stroke-related events, n (%) | 17 (5.8) |
| Acute stroke, n (%) | 4 (1.4) |
| New onset AF, n (%) | 14 (4.7) |
| Total | 29 (9.8) |

(SCD, sudden cardiac death; HF, heart failure; AF, atrial fibrillation).

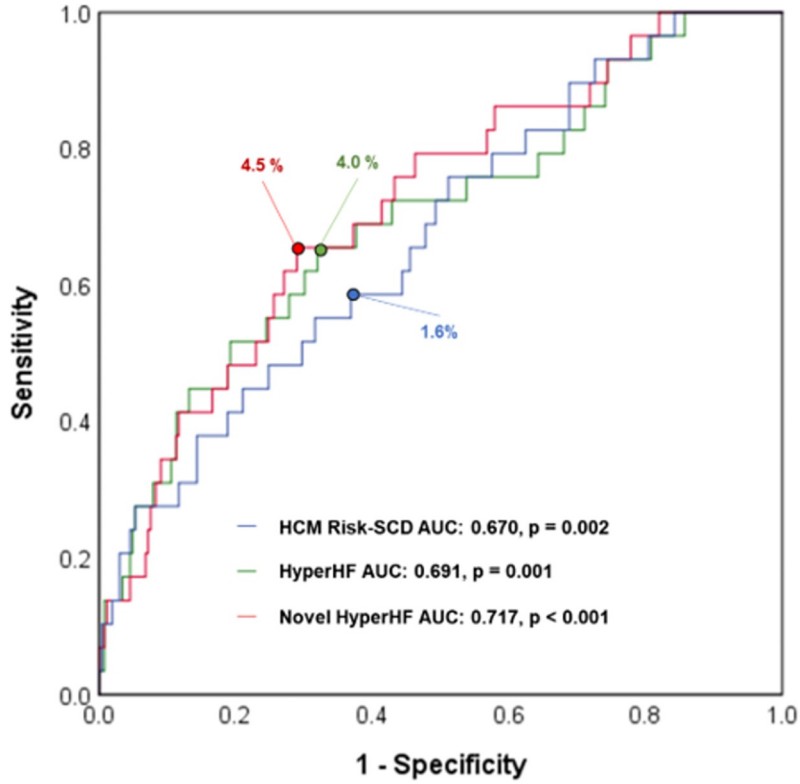

**Fig 2. Receiver operating characteristic curve of HCM Risk-SCD score, HyperHF score, and Novel HyperHF score for predicting overall events.** Prognostic ability of models for overall events was compared among HCM Risk-SCD, HyperHF score, and Novel HyperHF score. Novel HyperHF score showed the numerically highest AUC (0.670 vs. 0.697 vs. 0.717). The optimal cut off for predicting major adverse cardiac events was based on the receiver operating characteristic curve. Each number in the curve is a cut off. (ROC, receiver operating characteristic; AUC, area under the curve; HCM, hypertrophic cardiomyopathy; SCD, sudden cardiac death).

be applied in the prediction of overall risk of both SCD-related and stroke-related events [8,9,17]. Additionally, we proposed a novel risk prediction model to improve prediction of overall events in HCM.

## Prediction of SCD Riskin HCM

In HCM patients, SCD risk prediction is important to decide whether the patient should receive an implantable cardioverter defibrillator (ICD) therapy. The HCM Risk-SCD score was the most widely used method to predict the risk of SCD and guide the decision of ICD in clinical practice. ICD implantation typically is recommended for secondary prevention; it is used for the primary prevention when the 5-year risk score is higher than 4% by HCM Risk-SCD score method [9,18]. This study compared HyperHF and Novel HyperHF methods with HCM Risk-SCD. As a result, the prediction power of HCM Risk-SCD for overall events was lower than that of HyperHF and Novel HyperHF (AUC values 0.670, 0.691, and 0.717, respectively).

## Benefit of using CPET in HCM for risk prediction

In recent reports, exercise parameters on CPET such as peak oxygen consumption, minute ventilation to $CO_2$ production, and ventilatory anaerobic threshold predicted death from HF

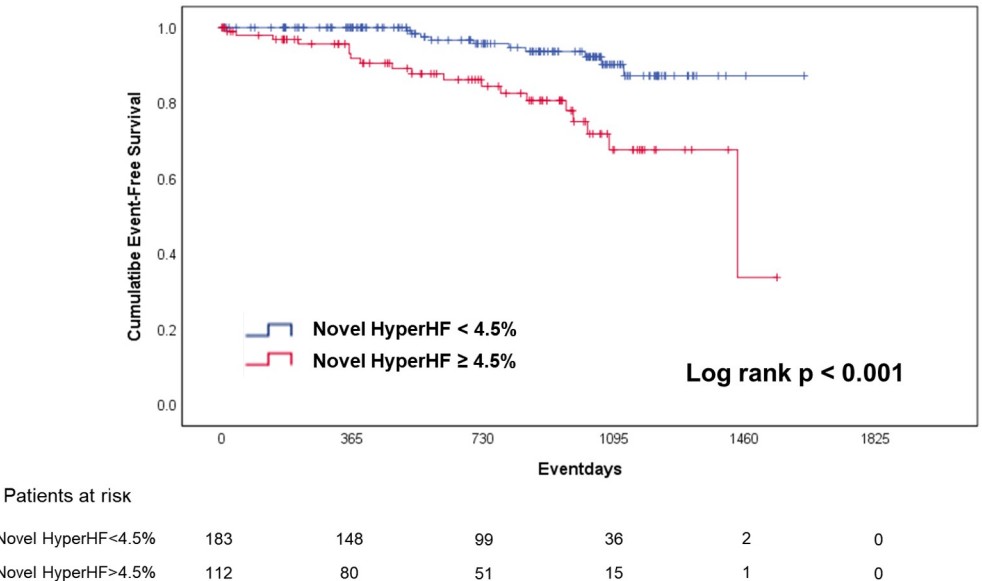

**Fig 3. Kaplan-Meier curve for event-free survival by Novel HyperHF score predicting overall events.** The cumulative event-free survival was compared between Novel HyperHF score ≥4.5% and <4.5% among total patients. Patients with higher Novel HyperHF score showed significantly lower survival after overall events during the follow-up period.

in HCM patients. Furthermore, CPET provides objective data regarding mechanisms and severity of functional limitation, which can further be applied to decision making for heart transplantation [10].

Previous HCM risk-SCD score was limited due to low sensitivity for clinically relevant decisions about ICD placement and might underestimate the number of high-risk patients who would remain unprotected and susceptible to sudden death without ICD therapy [4,19–23]. Application of CPET in patients with HCM can measure exercise function and prognosis in a variety of subsets of heart failure, including HCM. Substantial data has been collected showing that CPET is not only safe, but also a key element in comprehensive evaluation of HCM patients [24].

HyperHF score is combinational risk predicting model that integrates conventional echo-cardiological parameter (LAD) and CPET data such as $VE/VCO_2$ slope and circulatory power. HyperHF score is independently related to development of heart failure complications as well [25]. However, atrial fibrillation and stroke related risks exist in HCM which can consequently

**Table 5. Univariate and multivariate analysis of Novel HyperHF score and major risk factors predicting outcome.**

|  | Univariate | | Multivariate | |
|---|---|---|---|---|
|  | **P-value** | **CI** | **P-value** | **CI** |
| **Novel HyperHF** | <0.001 | 3.024–32.191 | 0.017 | 1.469–46.580 |
| **LAVI** | 0.014 | 1.016–1.149 | 0.266 | 0.987–1.049 |
| **LAD** | <0.001 | 1.014–1.049 | 0.698 | 0.938–1.101 |
| **Gender** | 0.519 | 0.302–1.829 | 0.056 | 0.107–1.029 |

(LAD, left atrial dimension; LAVI, left atrial volume index).

deteriorate patient's quality of life and survival. Unfortunately, there is no available scoring system that provides integrated risk prediction of both SCD-related and stroke-related events.

## Novel risk prediction model for cardio-cerebral outcome in HCM

With utilization of exercise parameters and echocardiographic parameters, Novel HyperHF score could provide better prediction of overall events related to HCM. Although assessment of LA enlargement appears to provide important information regarding stroke-related events, unidimensional M-mode LA antero-posterior diameter has limitations in representing true LA size [24]. LAVI could be a more sensitive markers for detecting the risk of clinical events in patients with HCM [26]. LAVI is associated with new-onset AF and stroke recurrence in embolic stroke of undetermined source patients and may be a better surrogate of atrial cardiopathy [27]. LAVI was also superior to LAD as an independent prognostic implication in terms of ischemic cardiomyopathy [28]. The mean and standard deviation were $41.2 \pm 6.0$ mm for the LAD and $36.7 \pm 13.9$ mL/m$^2$ for the LAVI, suggesting greater discrimination power of LAVI than LAD. Replacing LAVI in the formula of HyperHF score, Novel HyperHF score was the only statistically significant model for stroke-related event prediction.

Although LAVI alone is an important predictor for stroke in HCM, multivariate analysis suggested better prediction of both SCD and stroke-related events by Novel HyperHF score in our patient population.

## Clinical applications and perspectives of Novel HyperHF score

Both SCD-related risk and stroke-related risk exist in HCM and clinicians should monitor each risk in those patients who have diagnosed with the disease. Traditional risk models are not sufficient to predict both risks and we have created Novel HyperHF score to overcome limitations. Novel HyperHF score $\geq 4.5\%$ predicted significantly poor outcome including both cardio-cerebral outcome in HCM patients. More intensive diagnostic and follow up strategies need to be applied in those patients with higher Novel Hyper-HF score. Further validation study could broaden our perspectives.

## Study limitations

This study contains several limitations. First, this study was retrospective in design, causing inherent potential limitations. Second, atrial fibrillation, which is the most prevalent arrhythmia reaching 22.5% of HCM patients, was excluded from the present study to evaluate for the true prediction ability of Novel HyperHF score in future AF risk. Third, patients in a relatively healthy condition with a capability of CPET were included, and data from only HCM patients with CPET can be assessed with HyperHF or Novel HyperHF scoring system.

## Conclusions

For the overall risk prediction of cardio-cerebral outcome in HCM patients, previous HCM risk-SCD score and CPET-derived HyperHF score both provided fair prediction in this cohort from two tertiary University hospitals. Novel HyperHF substituted LAD for LAVI and showed better prediction of overall events in HCM. Novel HyperHF score might allow early identification of patients at high risk of SCD-related and stroke-related events. A future validation study using Novel HyperHF will further increase the impact of this new scoring system.

## Supporting information

**S1 Fig. Receiver operating characteristic curve of HCM Risk-SCD score, HyperHF score, and Novel HyperHF score for predicting SCD-Related events.** Prognostic ability of models for SCD-related events was compared among HCM Risk-SCD, HyperHF score, and Novel HyperHF score. HCM Risk-SCD score showed the highest AUC, while those of HyperHF score and Novel HyperHF were similar (0.715 vs. 0.695 vs. 0.692). The optimal cut off threshold for predicting major adverse cardiac events was based on the receiver operating characteristic curve. Each number in the curve is a cut-off value. (ROC, receiver operating characteristic; AUC, area under the curve; HCM, hypertrophic cardiomyopathy; SCD, sudden cardiac death). (TIF)

**S2 Fig. Receiver operating characteristic curve of HCM Risk-SCD score, HyperHF score, and Novel HyperHF score for predicting stroke-Related events.** Prognostic ability of models for stroke-related events was compared among HCM Risk-SCD, HyperHF score, and Novel HyperHF score. Novel HyperHF score showed the numerically highest AUC (0.607 vs. 0.620 vs. 0.659) and had only one p value less than 0.05 (0.138 vs. 0.098 vs. 0.028). The optimal cut off for predicting major adverse cardiac events was based on the receiver operating characteristic curve. Each number in the curve is a cut off value. (ROC, receiver operating characteristic; AUC, area under the curve; HCM, hypertrophic cardiomyopathy; SCD, sudden cardiac death). (TIF)

**S3 Fig. Kaplan-Meier graph of both Novel HyperHF score and HyperHF score.** The cumulative event-free survival was compared between Novel HyperHF score and HyperHF score among total patients. Patients with higher Novel HyperHF score showed significantly lower survival after overall events during the follow-up period than patients who had higher HyperHF score. (HCM, hypertrophic cardiomyopathy; SCD, sudden cardiac death). (TIF)

**S1 Table. Sensitivity and specificity, PPV, NPV of each cut-off point for the overall events.** (SCD, Sudden cardiac death; HCM, hypertrophic cardiomyopathy; PPV = positive predictive value; NPV = negative predictive value). (DOCX)

**S2 Table. Sensitivity and specificity, PPV, NPV of each cut-off point for the SCD-related events.** (SCD, Sudden cardiac death; HCM, hypertrophic cardiomyopathy; PPV = positive predictive value; NPV = negative predictive value). (DOCX)

**S3 Table. Sensitivity and specificity, PPV, NPV of each cut-off point for the stroke-related events.** (SCD, Sudden cardiac death; HCM, hypertrophic cardiomyopathy; PPV = positive predictive value; NPV = negative predictive value). (DOCX)

**S1 Graphical abstract.** (TIF)

## Author Contributions

**Conceptualization:** Jae-Man Lee, In-Cheol Kim, Ji-Hun Song, Jaewon Oh, Geu-Ru Hong, Seok-Min Kang.

**Data curation:** Jae-Man Lee, In-Cheol Kim, Hyungseop Kim, Jaewon Oh, Jong-Chan Youn, Geu-Ru Hong, Seok-Min Kang.

**Formal analysis:** Jae-Man Lee, Hyun-Bin Park, In-Cheol Kim, Ji-Hun Song.

**Funding acquisition:** Ji-Hun Song.

**Investigation:** Jae-Man Lee.

**Methodology:** Jae-Man Lee, Hyun-Bin Park, In-Cheol Kim, Jaewon Oh, Jong-Chan Youn, Geu-Ru Hong.

**Project administration:** Jae-Man Lee, Hyun-Bin Park, Jin-Eun Song, In-Cheol Kim, Ji-Hun Song, Jong-Chan Youn, Seok-Min Kang.

**Resources:** Jae-Man Lee, Hyun-Bin Park, In-Cheol Kim, Hyungseop Kim, Jong-Chan Youn, Geu-Ru Hong, Seok-Min Kang.

**Software:** Jae-Man Lee, Hyun-Bin Park, In-Cheol Kim, Geu-Ru Hong.

**Supervision:** In-Cheol Kim, Hyungseop Kim, Jong-Chan Youn, Geu-Ru Hong, Seok-Min Kang.

**Validation:** In-Cheol Kim, Jaewon Oh.

**Visualization:** In-Cheol Kim.

**Writing – original draft:** Jae-Man Lee, Hyun-Bin Park, Jin-Eun Song.

**Writing – review & editing:** Jin-Eun Song, In-Cheol Kim.

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
