## [Decision Letter · Decision Letter 0]

10 Mar 2021

PONE-D-21-01943

The Impact of Cardiopulmonary Exercise Derived Scoring on Prediction of Cardio-cerebral Outcome in Hypertrophic Cardiomyopathy

PLOS ONE

Dear Dr. Kim,

Thank you for submitting your manuscript to PLOS ONE. After careful consideration, we feel that it has merit but does not fully meet PLOS ONE’s publication criteria as it currently stands. Therefore, we invite you to submit a revised version of the manuscript that addresses the points raised during the review process.

We look forward to receiving your revised manuscript.

Kind regards,

Otavio R. Coelho-Filho, M.D., Ph.D., M.P.H.

Academic Editor

PLOS ONE

Journal Requirements:

"This research was supported by the Bisa Research Grant of Keimyung

University in 2017"

"The authors received no specific funding for this work."

Additional Editor Comments:

The current study investigated the prognostic role of cardiopulmonary exercise testing (CPET) to predict cardio-cerebral events in HCM patients.

Although the study may add relevant data to the current literature, reviewers have identified several issues requiring careful revision.

Additional comments:

1. Replace LV and LA dimensions by indexed volumes.

2. It is not clear whether echocardiographic strain data was available.

3. Since myocardial scar by Cardiac MRI has been shown to predict Cv events in HCM patients. clarify if Cardiac MRI data is available.

4. Detail information about CV events were not provided. It is unclear how SCD and CV were defend and confirmed.

5. Statistical analysis requires improvements.

6. In order to compare models in the current study I would suggest using Harrell’s C statistics to verify discrimination of risk prediction of models. Also Continuous Net Reclassification Index (NRI) and Integrated Discrimination Index (IDI) should be considered.

Reviewers' comments:

Reviewer's Responses to Questions

**Comments to the Author**

1. Is the manuscript technically sound, and do the data support the conclusions?

Reviewer #1: Partly

Reviewer #2: Yes

2. Has the statistical analysis been performed appropriately and rigorously? 

Reviewer #1: I Don't Know

Reviewer #2: Yes

3. Have the authors made all data underlying the findings in their manuscript fully available?

Reviewer #1: Yes

Reviewer #2: Yes

4. Is the manuscript presented in an intelligible fashion and written in standard English?

Reviewer #1: Yes

Reviewer #2: Yes

5. Review Comments to the Author

Reviewer #1: Major Comments

In the present study, the aim was evaluating a scoring system predicting cardio-cerebral events in Hypertrophic Cardiomyopathy (HCM) patients using cardiopulmonary exercise testing (CPET). A total of 295 consecutive HCM patients (age 59.9±13.2, 71.2 % male) who underwent CPET was included in the present study. The previous model for SCD risk score showed fair prediction ability. However, the prediction power of Novel HyperHF showed the highest value among the models. So, the authors concluded that both conventional HCM Risk-SCD score and CPET-derived HyperHF score were useful for prediction of overall risk of SCD-related and stroke-related events in HCM. However, a novel HyperHF score using LAVI could be utilized for a better prediction power.

Despite the relevance of the issue, the manuscript needs to be improved. The Introduction Section, Materials and Methods and Discussion Section should be significantly improved. The central aim of the present study is the importance of CPET in the risk stratification model to predicting cardio-cerebral events in HCM. However, the major conclusion is that “HCM Risk-SCD score and CPET-derived HyperHF score were useful for prediction of overall risk of SCD-related and stroke-related events in HCM… a novel HyperHF score using LAVI could be utilized for a better prediction power”

Minor Comments

Introduction Section

1- The authors should include more details about the possible role of CPET assessment in stratifying overall HCM prognosis, as well as HYPertrophic Exercise-derived Risk HF (HyperHF).

2- The authors should include that the aim of the present study is about risk stratification model in HCM patients

Materials and Methods Section

1- The authors should include more details about the clinical characteristics of HCM patients. Were obstructive HCM patients excluded?

2- The authors should include more details about cardiopulmonary exercise testing and related parameters. Type of cycle ergometer, protocol, etc

3- The authors should include HCM Risk-SCD model in the Risk Model Verification Section

Results Section

1- The authors should describe all the significantly results presented in Table 2 and Table 3.

Discussion Section

1- What is the importance to include CPET in the risk stratification model to predicting cardio-cerebral events in HCM? This is the central aim of the present study, and the authors should improve the discuss this issue.

2- In addition, the authors should improve the discuss about the Novel HyperHF using LAVI rather than LAD.

Reviewer #2: I have read with interest your work and it surely presents new data and provide further knowledge regarding CPET and HCM.

I do have some comments and suggestions:

- Although those predictive scores are quite new and might add new prognostic information, it demands additional data. In order to investigate whether a new risk scoring model would provide complementary prognostic information to previous echocardiographic variables and/or previous risk scores, a multivariable model should be included in the result part of the paper describing the scores, differences and demonstrating this better power prediction of the new score. In this matter:

a) A kaplan-Meyer curve with both scores at the same picture with differences on prediction.

b) A table with Hazard Ratios and p values comparing both scores and some key clinical variables such as atrial diameter and volume

c) A table with the incremental value of the new score using C-Statistics

- I wonder if authors could provide additional data regarding MRI and 24-hour ECG monitoring. Some important prognostic parameters such as fibrosis and NSVT during 24-monitoring are lacking.

- It seems that LA volume index is superior to LAD and might be an explanation for the additional predictive model of the new score. It is important to demonstrate that the new score itself is the responsible for the better accuracy not LA volume index alone - another multivariable model should be enough to assess this matter.

- EOV is a strong CPET parameter. Do you have data on this ?

- Some results are included in the discussion session. It should be included in the results session and the discussion should be addressed deeply on the differences between the findings among the scores and clinical characteristics.

6. PLOS authors have the option to publish the peer review history of their article (what does this mean?). If published, this will include your full peer review and any attached files.

Reviewer #1: No

Reviewer #2: No

---

## [Author Response · Author response to Decision Letter 0]

13 Jun 2021

I attached Rebuttal letter to reviewers. all my answers are in there. below scripts are manuscripts of file.

#Reviewer 1

Major Comments

In the present study, the aim was evaluating a scoring system predicting cardio-cerebral events in Hypertrophic Cardiomyopathy (HCM) patients using cardiopulmonary exercise testing (CPET). A total of 295 consecutive HCM patients (age 59.9±13.2, 71.2 % male) who underwent CPET was included in the present study. The previous model for SCD risk score showed fair prediction ability. However, the prediction power of Novel HyperHF showed the highest value among the models. So, the authors concluded that both conventional HCM Risk-SCD score and CPET-derived HyperHF score were useful for prediction of overall risk of SCD-related and stroke-related events in HCM. However, a novel HyperHF score using LAVI could be utilized for a better prediction power. 

Despite the relevance of the issue, the manuscript needs to be improved. The Introduction Section, Materials and Methods and Discussion Section should be significantly improved. The central aim of the present study is the importance of CPET in the risk stratification model to predicting cardio-cerebral events in HCM. However, the major conclusion is that “HCM Risk-SCD score and CPET-derived HyperHF score were useful for prediction of overall risk of SCD-related and stroke-related events in HCM… a novel HyperHF score using LAVI could be utilized for a better prediction power

#Answer

Thank you for your attentive comments. I totally agree that central aim and major aim is mismatch. Actual aim of our script is comparison of scoring models and prove of usefulness of CPET as a method of predicting prognosis. 

And for improvement of scripts, by CPET included as risk stratification method in 2020 AHA HCMP Guideline, I could add possible roles of CPET and HyperHF as risk stratification. And also added more details of study population, baseline characteristics, methods of CPET and details about HCM Risk-SCD model. Moreover, improve the importance of using not only CPET data as a risk stratification models but also LAVI than LAD. And for confirmative comparison of models, We proceeded Delong’s AUC comparison methods and added the results. 

Minor Comments

Introduction Section

# Comment 1

The authors should include more details about the possible role of CPET assessment in stratifying overall HCM prognosis, as well as HYPertrophic Exercise-derived Risk HF (HyperHF).

# Answer 1.

Thank you for your thorough review of the script. In recent 2020 AHA Hcmp Guideline, CPET has been inserted at risk stratification. It performed to quantify the degree of functional limitation and aid in selection of patients for heart transplantation or mechanical circulatory support. And it can be helpful in differentiating HCM from other causes of ventricular hypertrophy. And it also provides noninvasive method for assessing the cardiovascular, pulmonary and skeletal muscle components of exercise performance. (1)

And the HYPERHF score represents the first attempt of an integrated risk prediction model potentially expandable to generating individualized risk estimates for HFrelated events in a contemporary HCM population. And also CPET is useful in the evaluation of HCM patients. In this context, the HYPERHF score might allow early identification of those patients at high risk of HF progression and its complications.(2, 3)

#Comment 2

The authors should include that the aim of the present study is about risk stratification model in HCM patients

# Answer 2

Thank you for your detailed comment. As you suggest, my script has vague aim. We make amends for clarify that risk stratification and benefits of CPET are our aim.

Materials and Methods Section

# Comment 1

The authors should include more details about the clinical characteristics of HCM patients. Were obstructive HCM patients excluded? 

# Answer 1

Thank you for the detailed comment. We included 56 obstructive HCM patients, 113 non-obstructive patients and 126 Apical HCM patients. 

Included subtype numbers of HCM are Obstructive 56 patients, Non-obstructive 113 patients, Apical 126 patients.

# Comment 2

The authors should include more details about cardiopulmonary exercise testing and related parameters. Type of cycle ergometer, protocol, etc

# Answer 2

We totally agree with your comment. We added details about CPET and related parameters.

# Comment 3

The authors should include HCM Risk-SCD model in the Risk Model Verification Section

# Answer 3

Thank you for the advisory comment. I added HCM Risk-SCD model at Model Verification.

Results Section

# Comment 1

The authors should describe all the significantly results presented in Table 2 and Table 3.

# Answer 1

Thank you for valuable advice. I added all of siginifiant results.

Discussion Section

# Comment 1

What is the importance to include CPET in the risk stratification model in HCM? This is the central aim of the present study, and the authors should improve the discuss this issue. 

# Answer 1

We appreciate for your insightful comment. In 2020 AHA Guideline, CPET data such as Peak oxygen consumption and CO2 production predict death and provides objective data on the severity and mechanism for functional limitation. By adding these data, We can makes risk stratification model more accurate.

# Comment 2

In addition, the authors should improve the discuss about the Novel HyperHF using LAVI 

rather than LAD. 

# Answer 2

Thank you for thorough comments. I added benefits of using LAVI than LAD. 

Reviewer #2

 I have read with interest your work and it surely presents new data and provide further knowledge regarding CPET and HCM.

I do have some comments and suggestions:

- Although those predictive scores are quite new and might add new prognostic information, it demands additional data. In order to investigate whether a new risk scoring model would provide complementary prognostic information to previous echocardiographic variables and/or previous risk scores, a multivariable model should be included in the result part of the paper describing the scores, differences and demonstrating this better power prediction of the new score. In this matter:

a) A kaplan-Meyer curve with both scores at the same picture with differences on prediction.

b) A table with Hazard Ratios and p values comparing both scores and some key clinical variables such as atrial diameter and volume

c) A table with the incremental value of the new score using C-Statistics

Answer A):

Thank you for thoughtful comment. We make new graph for comparison. Hyper HF score seems devided well but Novel HyperHF score devided better.

Answer B): thank you for your thoughtful comments. I added the tables as supplement for compare. 

 Hazard Ratio P-value

Novel Hyper HF 3.654 0.005

Hyper HF 3.035 0.011

LAVI 1.031 <0.001

LAD 1.080 0.014

PeakVO2 0.905 <0.001

VE/VCO2 1.150 <0.001

HCMRisk-SCD 1.742 0.149

Answer C)

1.- I wonder if authors could provide additional data regarding MRI and 24-hour ECG monitoring. Some important prognostic parameters such as fibrosis and NSVT during 24-monitoring are lacking.

Answer: Thank you for your thoughtful comments. We can provide NSVT, Familiar history of SCD, Unexplained syncope history as a data parts of HCM-RiskSCD score. But Unfortunately We don’t have all patients’ data of MRI. But I agree that MRI data like fibrosis could be great prognosis factor. After much of data are collected I’ll proceed the study. 

2.- It seems that LA volume index is superior to LAD and might be an explanation for the additional predictive model of the new score. It is important to demonstrate that the new score itself is the responsible for the better accuracy not LA volume index alone - another multivariable model should be enough to assess this matter.

Answer: 

I appreciate to the reviewer for bringing out an important issue. We performed multivariable model analysis. As result, Novel hyperHF model is the responsible for the better accuracy itself. Here is results tables.

 Univariate Multivariate

 P-value CI P-value CI

Novel HyperHF <0.001 3.024-32.191 0.017 1.469-46.580

LAVI 0.014 1.016-1.149 0.266 0.987-1.049

LAD <0.001 1.014-1.049 0.698 0.938-1.101

Gender 0.519 0.302-1.829 0.056 0.107-1.029

And by inspired your advice, we performed Delong(1988) AUC comparison analysis for the new score. As a result it has no better accuracy than other scores. But in storke-related events, only p-value of AUC of Novel HyperHF below 0.05 so still in Stroke-related events new parameters cloud be used (Novel HyperHF score AUC0.659, p=0.028; Hyper HF AUC 0.620, p=0.098; HCM Risk-SCD AUC 0.607, p=0.138). Results of Delong AUC comprarisons are at below graph.

Cc 　 　 　

 HCM Risk-SCD score HyperHF score Novel HyperHF score

HCM Risk-SCD score 　 0.6814 0.4479

HyperHF score 　 　 0.2017

Novel HyperHF score 　 　 　

SCD related risk 　 　 　

 HCM Risk-SCD score HyperHF score Novel HyperHF score

HCM Risk-SCD score 　 0.8335 0.7990

HyperHF score 　 　 0.7904

Novel HyperHF score 　 　 　

Stroke related risk 　 　 　

 HCM Risk-SCD score HyperHF score Novel HyperHF score

HCM Risk-SCD score 　 0.9843 0.6225

HyperHF score 　 　 0.0913

Novel HyperHF score 　 　 　

3.- EOV is a strong CPET parameter. Do you have data on this ?

Answer: Thank you for your valuable advice. We know there is a research that EOV could be great prognostic factor for Hypertrophic cardiomyopathy like peakVO2 and VE/VCO2. (27). Unfortunately this study is retrospective and we do not checked EOV. In next time, Using EOV as a part of predicting model might be interesting and could be better choice. 

4.- Some results are included in the discussion session. It should be included in the results session and the discussion should be addressed deeply on the differences between the findings among the scores and clinical characteristics.

Answer): I totally agree with your advice. We revised results session and added discussion about differences between the findings among the scores and clinical characteristics.

---

## [Decision Letter · Decision Letter 1]

29 Jul 2021

PONE-D-21-01943R1

The Impact of Cardiopulmonary Exercise Derived Scoring on Prediction of Cardio-cerebral Outcome in Hypertrophic Cardiomyopathy

PLOS ONE

Dear Dr. Kim,

Thank you for submitting your manuscript to PLOS ONE. After careful consideration, we feel that it has merit but does not fully meet PLOS ONE’s publication criteria as it currently stands. Therefore, we invite you to submit a revised version of the manuscript that addresses the points raised during the review process.

We look forward to receiving your revised manuscript.

Kind regards,

Otavio R. Coelho-Filho, M.D., Ph.D., M.P.H.

Academic Editor

PLOS ONE

Journal Requirements:

Additional Editor Comments (if provided):

Authors were able to address all comments.

The current manuscript have significantly improved and may merit publication.

I believe that the current manuscript may improve its readability if revised by a native English speaker.

I have advised the authors to have a native English speaker to proofread the manuscript.

Reviewers' comments:

Reviewer's Responses to Questions

**Comments to the Author**

1. If the authors have adequately addressed your comments raised in a previous round of review and you feel that this manuscript is now acceptable for publication, you may indicate that here to bypass the “Comments to the Author” section, enter your conflict of interest statement in the “Confidential to Editor” section, and submit your "Accept" recommendation.

Reviewer #1: All comments have been addressed

Reviewer #2: All comments have been addressed

2. Is the manuscript technically sound, and do the data support the conclusions?

Reviewer #1: Yes

Reviewer #2: Yes

3. Has the statistical analysis been performed appropriately and rigorously? 

Reviewer #1: I Don't Know

Reviewer #2: Yes

4. Have the authors made all data underlying the findings in their manuscript fully available?

Reviewer #1: Yes

Reviewer #2: Yes

5. Is the manuscript presented in an intelligible fashion and written in standard English?

Reviewer #1: Yes

Reviewer #2: Yes

6. Review Comments to the Author

Reviewer #1: (No Response)

Reviewer #2: (No Response)

7. PLOS authors have the option to publish the peer review history of their article (what does this mean?). If published, this will include your full peer review and any attached files.

Reviewer #1: No

Reviewer #2: No

---

## [Author Response · Author response to Decision Letter 1]

12 Sep 2021

we include amended statements within response to reviewer rivison2. below is the script.

#Dear Otavio R. Coelho-Filho.

Thanks a lot for revising and comments to our manuscripts. Below sentences are major requirements of minor revision.

Before start, I revised authors one more time with their consents. If I need any paper, notice me please.

# Comment 1.

# Answer 1.

 Thank you for comments. I separated letters which for editor, named Response to Reviewers.

# Comment 2

# Answer 2.

I made Revised manuscript with track changes with yellow highlights. 

# Comment 3.

# Answer 3.

I attached manuscript file. Thank you.

# Comment 4.

Please review your reference list to ensure that it is complete and correct. If you have cited papers that have been retracted, please include the rationale for doing so in the manuscript text, or remove these references and replace them with relevant current references. Any changes to the reference list should be mentioned in the rebuttal letter that accompanies your revised manuscript. If you need to cite a retracted article, indicate the article’s retracted status in the References list and include a citation and full reference for the retraction notice. 

# Answer 4.

I revised Response to reviewer files and erased duplicated references. 

Thank you for critical comments and I revised it. If there is retracted reference that I missed and I committed mistakes, then please give us comments. 

And in addition, 19th reference is too old guidelines (2011 year) so I change it as new one (2020 guidelines)s

# Comment 5.

I believe that the current manuscript may improve its readability if revised by a native English speaker. I have advised the authors to have a native English speaker to proofread the manuscripts.

# Answer 5.

Thank you for thoughtful comments. We commissioned our manuscripts English Correction. 

# Comment 6.

# Answer 6.

Thank you for informing us great methods for revising figures. I completed revising figures by using mentioned method. 

Sincerely, Jaeman Lee, MD.

---

## [Editor Report · Decision Letter 2]

25 Oct 2021

The Impact of Cardiopulmonary Exercise Derived Scoring on Prediction of Cardio-cerebral Outcome in Hypertrophic Cardiomyopathy

PONE-D-21-01943R2

Dear Dr. Kim,

We’re pleased to inform you that your manuscript has been judged scientifically suitable for publication and will be formally accepted for publication once it meets all outstanding technical requirements.

Kind regards,

Otavio R. Coelho-Filho, M.D., Ph.D., M.P.H.

Academic Editor

PLOS ONE

Additional Editor Comments (optional):

The authors satisfactory responded all questions.
---

## [Editor Report · Acceptance letter]

7 Jan 2022

PONE-D-21-01943R2 

The Impact of Cardiopulmonary Exercise-derived Scoring on Prediction of Cardio-cerebral Outcome in Hypertrophic Cardiomyopathy 

Dear Dr. Kim:

I'm pleased to inform you that your manuscript has been deemed suitable for publication in PLOS ONE. Congratulations! Your manuscript is now with our production department. 

Kind regards, 

on behalf of

Dr. Otavio R. Coelho-Filho 

Academic Editor

PLOS ONE